

# 1   Behavior of Saline Ice under Cyclic Flexural Loading

Andrii Murdza[1], Erland M. Schulson[1], Carl E. Renshaw[1,2]
[1]Thayer School of Engineering, Dartmouth College, Hanover, NH, USA, 03755
[2]Department of Earth Sciences, Dartmouth College, Hanover, NH, USA, 03755
*Correspondence to*: Andrii Murdza (Andrii.Murdza@dartmouth.edu)
**Abstract.** New systematic experiments reveal that the flexural strength of saline S2 columnar-grained ice loaded
normal to the columns can be increased upon cyclic loading by about a factor of 1.5. The experiments were conducted
using reversed cyclic loading over ranges of frequencies from 0.1 to 0.6 Hz and at a temperature of -10 ℃ on saline
ice of two salinities: 3.0±0.9 and 5.9±0.6 ‰. Acoustic emission hit rate during cycling increases with an increase of
stress amplitude of cycling. Flexural strength of saline ice of 3.0±0.9 ‰ salinity appears to increase linearly with
increasing stress amplitude, similar to the behavior of laboratory-grown freshwater ice (Murdza et al., 2020c) and to
the behavior of lake ice (Murdza et al., 2020a). The flexural strength of saline ice of 5.9±0.6 ‰ depends on the vertical
location of the sample within the thickness of an ice puck; i.e., the strength of the upper layers, which have a lower
brine content, was found to be as high as three times that of lower layers. Flexural strength is governed by tensile
strength which appears to be controlled by crack nucleation. Cyclic strengthening is attributed to the development of
an internal back stress that opposes the applied stress and originates possibly from dislocation pileups. The fatigue life
of saline ice is erratic.

## 18   1. Introduction

19         Fatigue of materials is a subject of great practical importance in engineering and has been widely studied.
Fatigue refers to changes in material properties resulting from cyclic loading. The fatigue strength of materials is
typically controlled by microcrack formation and subsequent growth that leads to weakening .

23         It is not surprising that fatigue appears to play an important role in sea ice mechanics. For example, the Arctic
and Antarctic floating ice covers and ice shelves are subjected to cyclic loading from ocean swells that can penetrate
deeply into an ice pack and potentially result in the breakup of the ice cover (Squire, 2007). Such events, where under
the action of surface waves a floating ice cover exhibited sudden breakup into smaller pieces, have been repeatedly
witnessed and described (Shackleton, 1982; Liu and others, 1988; Prinsenberg and Peterson, 2011; Asplin and others,
2012; Collins and others, 2015; Kohout and others, 2016; Hwang and others, 2017). Ice cover breakup leads to a
decline in the albedo (Pistone and others, 2014; Zhang and others, 2019) and to the intensification of melting. Also,
smaller ice floes attenuate ocean waves less than the parent solid ice cover, thereby endangering coastal zones to
erosion. Given the retreat of the sea ice cover and the attendant increase in oceanic fetch, larger waves are expected
to develop; correspondingly, the remaining ice cover is expected to be subjected to episodes of greater cyclic forcing.
The potential for fatigue failure is thus increasing.




Cyclic loading may also play an important role in other scenarios. For instance, during ice-structure
interactions (Jordaan, 2001; Hendrikse and Metrikine, 2016; O'Rourke and others, 2016; Jordaan and others, 2008)
the structure itself, such as a light-house, may be weakened to a degree that depends on the strength of the ice. Other
examples are runways and roads that are built by freezing water on cold oceans, rivers and lakes and subsequently
subject to cyclic loading. Therefore, it is important to understand the behavior of ice under cyclic loading.

Currently, the effects of cyclic loading on the physical and mechanical properties of sea ice and to the
susceptibility of the material to fatigue are poorly constrained. Tabata and Nohguchi (1980) conducted experiments
on sea ice sampled from Lake Saroma, Hokkaido, Japan and from Barrow, Alaska. They loaded the ice cyclically
under uniaxial compression between two specified stress levels under a variety of combinations of strain rate (from
$10^{-5}$ s$^{-1}$ to $10^{-2}$ s$^{-1}$), temperature (from -2 °C to -24 °C) and orientation (horizontal and vertical). They found that with
a decrease of average stress and with a decrease of amplitude, the time to failure increases; and by lowering the
temperature, the time to failure and the number of cycles also increases.

Other evidence of the weakening of sea ice under wave-driven in situ cyclic loading is discussed by Haskell
and others (1996), Bond and Langhorne (1997), Langhorne and others (1998), (1999), (2001). In these works the
authors obtained an S-N fatigue curve (S, upper peak stress of cycling – N, number of cycles imposed to failure),
typical of curves obtained from engineering materials, i.e. for lower stress amplitude more cycles are needed for
failure. The authors stated that the endurance limit, that is the stress amplitude below which the sea ice can withstand
an unlimited number of cycles, is approximately one-half the failure stress of non-cycled ice.

The constitutive behavior of saline ice under cyclic loading was also investigated previously (Cole, 1995,
1998; Cole et al., 1998, 2002; Cole and Dempsey, 2004; Cole and Durell, 1995; Dempsey et al., 2003; Wei et al.,
2020), specifically, inelastic deformation of sea ice was explored via dislocation-based mechanism. In these works
the authors investigated the effect of temperature (from -5 to -50 °C), microstructure (total porosity varied from 14 to
104 ppt), cyclic stress amplitude (from 0.04 to 0.8 MPa), loading frequency (from $10^{-3}$ to 1 Hz), dry isothermal vs
floating specimens on the response of the ice. However, the strength of ice after it had been cycled was not measured.

Nothing more (to our knowledge) has been reported on the fatigue of sea ice. The topic is absent from a
critical review by Squire (2007) and from two recent books on ice (Schulson and Duval, 2009; Weeks, 2010).

The behavior summarised above indicating the weakening of ice under cyclic loading, obtained from
experiments conducted on saline and sea ice, might possibly account for the sudden breakup of natural ice covers.
However, this behavior appears in conflict with the behavior of freshwater ice under cyclic loading (Cole, 1990; Gupta
et al., 1998; Iliescu et al., 2017; Iliescu and Schulson, 2002; Murdza et al., 2019, 2020c, 2020a). In those experiments,
it was discovered that the ice flexural strength increases upon repetitive loading, followed by recovery upon post-



cycling annealing (Murdza et al., 2020b). This differene in the behavior of the two kinds of ice could perhaps be
attributed to the presence of defects in sea/saline ice, such as brine pockets , brine channels and non-penetrating
microcracks. Such defects serve as stress concentrators, thereby lessening the need to nucleate cracks to the degree
that fatigue life may be governed primarily by crack propagation.

Therefore, given that limited information about the behavior of sea/saline ice under cycling, and given the
discrepancy in behavior of fresh and sea/saline ice, we conducted a study under controlled conditions in the laboratory
on the flexural behavior of saline ice . In this paper, we describe the experiments in which plates of S2 columnar-
grained saline ice of two salinities (3.0±0.9 and 5.9±0.6 ‰) were subjected at -10 ℃ to four-point, reverse cycling at
~0.1-0.6 Hz and then, after several hundred or more cycles, were bent to failure, provided the plates did not break
during cycling. We chose the rate of cycling to simulate the vibration frequency of a natural sea ice cover (Collins et
al., 2015).
**2. Experimental procedure**
**2.1 Ice growth and characterization**
We studied saline ice of two melt-water salinities: 3.0±0.9 and 5.9±0.6 ppt, where ± sign indicates standard
deviation. We produced the ice in the laboratory in a manner described previously (Golding et al., 2014). Briefly,
solutions containing 17.5 ± 0.2 ppt and 35 ± 0.2 ppt (parts per thousand, or ‰) of the commercial product "Instant
Ocean" salt mixture were prepared and then frozen unidirectionally downward over a period of about 7 days. This
produced pucks ~1 m in diameter and ~0.3 m thick. A bottom layer of ice of about 7-10 cm was discarded as it was
slushy and weak. Melt-water salinity was measured using a calibrated YSI Pro30 conductivity and salinity probe.

Figure 1 shows the microstructure of the ice and Table 1 lists its density and grain size; Figure 2 shows
stereographic projections of the crystallographic c-axes. The ice is characterized by columnar-shaped grains whose
growth texture is marked by c-axes confined within about 15° of the horizontal plane and randomly oriented in that
plane. In other words, the ice is termed S2, after Michel and Ramseier (1971), and is similar to natural first-year sea
ice (for comparison, see Figure 3.7 of Schulson and Duval (2009)). Grain size of S2 ice is the averge diameter of the
columnar-shaped grains.
**2.2. Growth features**
The ice contained both sub-mm sized brine pockets and supra-mm sized drainage channels, reminiscent of
natural sea ice. The ice of lower salinity (3.0±0.9 ppt) had fewer defects of both kinds. Some of the ice of higher
salinity (5.9±0.6 ppt) possessed channels whose size was almost as large as the grain diameter. The defects scattered
light to the degree that in bulk form the ice had an overall opaque appearance, while in thin section (~1mm) it exhibited
to the naked eye distinct linear whitish features which we took to be sets of interconnected brine pockets. The ice of
higher salinity possessed more of these features, especially near the bottom of the parent puck (which was the last part



to solidify). Figures 3 and 4 show examples. Our sense is that these features served as stress concentrators, particularly
ones that traversed the test specimen (described below), thereby weakening the ice. Indeed, as will become apparent
below, samples obtained from near the bottom of a puck of higher salinity (5.9±0.6 ppt) had relatively low flexural
strength.

Because the ice of both salinities exhibited a different visual appearance from the top and bottom of the parent

puck, in preparing test specimens for flexing we distinguished them by their position (depth) within an ice puck from
which they were prepared,


Table 2. As it turned out, however, distinction in terms of depth for the ice of lower salinity (and fewer

defects) did not  correlate in a systematic manner with the strength of the specimen when scatter in the data was taken
into account (more below). The flexural strength of ice plates prepared from an ice puck of higher salinity (and more
defects) appears to depend on the depth of ice from which ice plates were prepared, although we performed fewer
tests on the ice of higher salinity.

**2.3. Sample preparation and test setup**

Once the ice had been grown, it was cut into blocks of dimensions ~ 10 x 30 x 20 cm$^3$. The blocks were

stored in a cooler (at -10 °C) on their side (such that columnar-shaped grains were oriented horizontally) to reduce
brine drainage.

Specimens for flexing were manufactured from the ice blocks in the form of thin plates of dimensions

$h$ ~16 mm in thickness (parallel to the long axis of the grains), $b$ ~ 85 mm in width, and $l$ ~300 mm in length. The test
specimens were allowed to equilibrate to the test temperature of -10 °C for at least 24 hours before testing.

A detailed description of the specimens' preparation and loading can be found elsewhere (Iliescu et al., 2017;

Murdza et al., 2018, 2019, 2020c). To summarize: The ice plates were flexed up and down under 4-point loading
under constant displacement rate using a servo-hydraulic loading system (MTS model 810.14) to which we attached
a custom-built 4-point loading frame, Figure 5. A load cell, calibrated for both tension and compression, and a linear
variable differential transformer (LVDT) gauge were used for measurements of load and the displacement of the upper
surface of the ice plate during cycling. Acoustic emissions were recorded during cycling using a PCI-2 18-bit A/D
system; its frequency response is 3 kHz–3 MHz and its minimum AE amplitude detection threshold was set to 45 dB.
We used a micro 30STC sensor (9.5 mm diameter, 11 mm thickness) which was attached to the top surface of an ice
plate with a rubber band. Vacuum grease was used as the coupling agent between the sensor and the ice surface.

The experiments were performed in a cold room at a temperature of -10°C and at an outer-fiber center-point

displacement rate of 0.1 mm s$^{-1}$ (or outer-fiber strain rate of about 1.4 x 10$^{-4}$ s$^{-1}$). This displacement rate resulted in an
outer-fiber stress rate in the range from ~ 0.3 to 0.5 MPa s$^{-1}$ and frequencies in the range from 0.1 to 0.6 Hz (i.e.
periods from ~10 to 1.5 sec), which, as already noted, is similar to the frequency of ocean swells (Collins et al., 2015).
The major outer-fiber stress $\sigma_f$ was calculated from the relationship:

$$\sigma_f = \frac{3PL}{4bh^2},$$     (1)

where $P$ is the applied load and $L$ is the distance between the outer-pair of loading cylinders (shown in Figure 5b) and
is set by the geometry of the apparatus to be $L = 254$ mm.
Measurements of load and of displacement versus time at the beginning and near the end of cycling revealed

little evidence of softening during the tests, similar to the case for freshwater ice (Iliescu et al., 2017; Murdza et al.,
2020c).

We used two different loading procedures, as we did earlier in our study of S2 freshwater ice. Type I loading

was a completely reversed stress cycle with constant stress amplitude and mean stress of zero. Type II was similar to
Type I but incorporated an increasing multi-level (or step-level) stress amplitude. This second type of loading
essentially consisted of several Type I steps of increasing stress amplitudes. In the present study for stress amplitudes
below 0.7 MPa we used Type I loading. To cycle ice samples at stress amplitudes above 0.7 MPa, we first pre-
conditioned specimens through step-loading Type II procedure at progressively higher stress amplitude levels (see
Iliescu et al. (2017) and Murdza et al. (2018) for details). After pre-conditioning, samples were cyclically loaded
according Type I loading at least 300 times and generally for ~2000 times.
**3. Results and Observations**

In the present study failure of specimens during cycling occurred more frequently than in the study on

freshwater ice (Murdza et al., 2020c). Hence, the propensity for failure during cycling is greater in saline ice, owing
to the stress-concentrating effects of the brine pockets and channels noted above. Fatigue life of saline ice per se is
described below in Section 3.4.
**3.1. Flexural strength of non-cycled ice**

The flexural strength of non-cycled saline ice of both salinities was measured at -10 ℃ and at a nominal

outer-fiber center-point displacement of 0.1 mm s$^{-1}$. The results are listed in


Table 2. The average and standard deviation of the measured flexural strength of saline ice of lower salinity

(3.0±0.9 ppt) are 0.96±0.13 MPa. As mentioned above, the strength of the lower salinity ice did not correlate
systematically with the depth of the parent puck from which ice plates were prepared. The measured strength compares





favorably with the value of 0.85±0.20 MPa reported by Timco and O'Brien (1994) for sea ice of similar salinity, as
can be seen in Figure 6. Brine volume fraction $v_b$ was calculated according to Frankenstein and Garner (1967):

$$v_b = 0.001 * S \left( \frac{49.185}{|T|} + 0.532 \right),$$    (2)


where $T$ is temperature in degrees Celsius between -0.5 °C and -22.9 °C and $S$ is melt-water salinity (in ppt) of the ice.

The average and standard deviation of the measured flexural strength of saline ice of higher salinity

(5.9±0.6 ppt) are 0.98±0.36 MPa. The obtained values (Figure 6) deviate slightly towards higher values compared to
the data of Timco and O'Brien (1994), although scatter is significantly greater if compared with the ice of lower
salinity (3.0±0.9 ppt). This may be explained by the greater degree of interconnectivity of brine pockets at the bottom
of an ice puck (discussed above and shown in Figures 3 and 4) . This result shows how much the strength of ice is
sensitive to flaws and defects. Given that larger volumes usually contains larger defects, the flexural strength of sea
ice on the medium and larger scale of the field (Karulina et al., 2019) is lower than on the smaller scale of the
laboratory.

We also compare our measurements of flexural strength with the tensile strength of sea ice. For this purpose,

and as we did in our previous work on freshwater ice (Iliescu et al., 2017; Murdza et al., 2020c), flexural strength is
divided by 1.7 (Ashby and Jones, 2012), because the volume of the material which is subjected to the highest stress
in bending is smaller than in uniaxial tension; thus, the largest defect which governs the failure may not be near the
surface of a bent specimen. Upon dividing the flexural strength of the non-cycled saline ice of lower salinity by 1.7,
we found the average across-column tensile strength from our experiments to be 0.96±0.13 MPa/1.7 = 0.56±0.08 MPa.
This value compares favorably with the values  0.56±0.06 MPa and 0.63±0.12 MPa reported by Richter-Menge and
Jones (1993) for the tensile strength of columnar-grained first-year sea ice of 4.1±0.3 ppt salinity loaded uniaxially
across the columns at a temperature of -10 °C and strain rates of $10^{-5}$ and $10^{-3}$ s$^{-1}$. Recall that in the present experiments
the outer-fiber strain rate was about 1.4 x $10^{-4}$ s$^{-1}$ which is within the range reported by Richter-Menge and Jones
(1993). This agreement between direct and indirect measurements of tensile strength lends confidence that our lab-
grown saline ice is a reasonably faithful analogue of natural sea ice.
**3.2. Flexural strength versus number of reversed cycles under constant low stress amplitude**

To find whether there is a relationship in saline ice of lower salinity (3.0±0.9 ppt) between the flexural

strength and number of cycles imposed under a constant stress amplitude, we performed via Type-I loading a series
of experiments at -10 °C at an outer-fiber center-point displacement rate of 0.1 mm s$^{-1}$ at a low stress amplitude of
0.35 MPa; i.e., at an amplitude less than one-half the flexural strength of non-cycled ice. Figure 7 shows the results.
The number of cycles varied from about 100 to 14000. The average strength and standard deviation of all data from
Figure 7 are 0.96±0.23 MPa. For comparison, the strength and standard deviation of non-cycled ice are



0.96±0.13 MPa, which implies that no strengthening at cycling stress amplitude of 0.35 MPa occurs. For freshwater
ice (Murdza et al., 2020c), we found that once the number of cycles at a given low stress amplitude exceeded  300,
the number of cycles had no significant effect on the flexural strength, implying that a kind of saturation of strength
developed. Given that result and the new resuts for saline ice, we followed the practice in the present study of cycling
more than 300 times, often as many as 2000 times, before bending the ice to failure. We termed the strengths so
obtained the saturated strength.
**3.3. Flexural strength versus stress amplitude**

The (saturated) flexural strength increases with stress amplitude. Figure 8 shows measurements obtained

from saline ice of both salinities cycled at -10 °C at an outer-fiber displacement rate of 0.1 mm s$^{-1}$. For comparison,
data from laboratory grown freshwater ice (Murdza et al., 2020c) of S2 character and from lake ice of the same
character (Murdza et al., 2020a) are also shown. The relationship between the flexural strength, $\sigma_{fc}$ and cycled stress
amplitude, $\sigma_a$, for saline ice appears to be a linear one and, within experimental scatter, to have essentially the same
sensitivity to stress amplitude as freshwater ice; namely:

$$\sigma_{fc} = \sigma_{f0} + k\sigma_a \ , \qquad\qquad (2)$$


where $k = 0.68$ is a constant. For freshwater ice $\sigma_{f0} = 1.75$ MPa is the flexural strength of non-cycled ice. For the
saline ice $\sigma_{f0} = 0.96$ MPa. There is. perhaps, in Figure 8 a hint that for saline ice there is a threshold of about 0.4 MPa
that must be exceeded to detect strengthening. We refrain from putting too fine a point on this until more data become
available. Although saline ice is weaker than freshwater ice, it appears its strength increases at the same rate as
freshwater ice upon cycling under a given amplitude of the outer fiber stress.

The maximum degree of strengthening in the case of saline ice of lower salinity (3.0±0.9 ppt) is significantly

lower than that for the freshwater ice. Specifically, we were able to strengthen saline ice by about 50% of the non-
cycled strength compared with about 100% for freshwater ice (Murdza et al., 2020c). Another important point to
mention is that we almost were not able to cycle specimens at stress amplitudes greater than the flexural strength of
non-cycled material, whereas in the case of freshwater ice we were able to cycle at stress amplitudes significantly
greater than flexural strength of non-cycled ice. Indeed, the maximum cycled stress amplitude we were able to reach
in the case of saline ice of lower salinity (3.0±0.9 ppt) during all tests was 1.1 MPa, which is not statistically different
from the non-cycled flexural strength of 0.96±0.13 MPa**.**

For saline ice of lower salinity (3.0±0.9 ppt), there is no evidence that the flexural strength of both non-

cycled and cycled ice is significantly affected by the depth of ice from which ice plates were prepared. For saline ice
of higher salinity (5.9±0.6 ppt), however, the flexural strength of both non-cycled and cycled ice appears to depend
on the depth of ice from which ice plates were prepared, Figure 9. Indeed, the flexural strength of samples from the





bottom and from the top of an ice puck of higher salinity (5.9±0.6 ppt) differs by ~3 times (~0.4 MPa vs ~1.4 MPa).
More data from the ice of higher salinity are needed to be more specific on this point.

**3.4. Fatigue behavior**

The samples from which the data in Figure 8 were obtained did not fail during cycling; moreover, some of
them obtained statistically significant strengthening. However, a sufficient number of specimens failed while cycling
which allows us to construct S-N fatigue curve for the fatigue life of saline ice of lower salinity (3.0±0.9 ppt) at -10°C
and 0.1 mm s$^{-1}$ outer-fiber displacement rate. The S-N behavior is shown in Figure 10. The number of cycles here is
the number of cycles to failure during cycling at the last stress amplitude level and not the total number of cycles. The
S-N trend does not show any systematic dependence of the number of cycles to failure on stress amplitude. Indeed,
for the same stress amplitude of ~ 0.9 MPa, fatigue failure occurred after as few as <10 cycles and after as many as a
few thousand cycles. No systematic trend was also observed when stress amplitude plotted versus total number of
cycles. Statistical analyses to test the hypothesis that the slope in Figure 10 is zero resulted in a p-value equal ~0.06.
Therefore, there is no significant effect of number of cycles on the stress at which failure occurred at 5% significance
level. We attribute this variability in fatigue life to the variability in microstructure from specimen to specimen.
That said, a note of caution is appropriate. The data in Figure 10 should not be viewed as fatigue data in the
usual sense; i.e., in the way such data are viewed when obtained from other materials (e.g., metals and alloys) that
exhibit classical fatigue behavior. In those cases, before cycling, all specimens are assumed to have the same thermal-
mechanical history. That was not the case here for the saline ice, as most of the samples were pre-conditioned
according to Type II procedure before they were cycled at the last stress level where they failed while cycling. In other
words, in order to get fatigue failure, we were increasing stress amplitude by small increments of ~0.05 MPa and
allowed a sufficient number of cycles at each stress level (~500-1000) before we reached a fatigue failure.
The question to address here is why we did not obtain a classical S-N curve? We suggest that the classical
mechanism of fatigue, i.e. accumulation of damage, is not in play in our tests and some other process is controlling
fatigue life.

**3.5 Experimental observations of samples after fatigue failure**

Based on the process of classical fatigue behavior, fatigue life is governed by the accumulation of damage
and hence through the combination of crack nucleation and crack propagation. Even though the scale of the
microstructure of saline ice is relatively coarse in relation to the microstructure of most metals and alloys, owing to
the opacity of saline ice it is difficult to track by the unaided eye crack nucleation and subsequent growth during
cycling . However, we can look into the microstructure after fatigue failure using other methods such as optical
microscopy and thin-sections. In order to determine whether classical mechanism operates in our tests, we need to



show that there are remnant microcracks in the middle part of our sample. The observations described below were
obtained from saline ice of lower salinity (3.0±0.9 ppt).

After fatigue failure occurred during testing, test samples were observed in the optical microscope with
magnification of up to 50x in order to look for newly formed remnant microcracks. In the vast majority of samples
we did not find any evidence of new damage; only in a few samples we were barely able to detect one or two
microcracks after cycling.

A few samples that failed in fatigue were examined by thin-section analysis. Three thin sections were
prepared from every specimen (four specimens were investigated) in order to ensure a greater probability of observing
microcracks growing from brine pockets or brine channels, should they be present. The plane of the thin section was
parallel to the long axis of the columnar grains and parallel to the direction of the greater normal stress. This plane
was taken as the best plane to observe possible cracks. Thin sections were observed using non-polarized light. We
found no evidence of microcracks starting their growth from brine pockets or from other defects. In fact, we found no
microcracks at all.

Thus, we suggest that once one microcrack was nucleated or once the stress reached a critical value at one of
the brine pockets or other defects, the crack propagated immediately through the thickness of sample. Based on this
observation, it appears that slow crack growth is not a significant contribution to the fatigue life of the plates of the
laboratory-grown saline ice that we studied.
**3.6. Acoustic emissions**
Acoustic emissions (AE) during repetitive loading of ice have been previously recorded and analyzed in
laboratory and in situ (Langhorne and Haskell, 1996), (Cole and Dempsey, 2006, 2004; Lishman et al., 2020; Murdza
et al., 2020c). Langhorne and Haskell (1996) suggested that the emissions originate either from dislocation breakaway
or from microcracking associated with dislocation motion.

In contrast to freshwater ice, where no sound was detected until failure (Murdza et al., 2020c), continuous
emission was detected while cycling at constant stress amplitude. Figure 11 shows the cummulative hits as a function
of time for ice that was cycled reversely at a constant stress amplitude of 0.5 MPa. As can be seen, the rate of hits (or
hits per unit time), which is a slope of the curve in Figure 11, is about the same for the duration of the experiment.

Interestingly, the hit rate depends on stress amplitude during cycling. Figure 12 shows this behavior . The
greater is the stress amplitude, the greater is the hit rate. However, during cycling below about 0.2 MPa no AE were
detected.



Figure 12 also indicates that the hit rate is independent of the sequence of different stress amplitudes. The
numbers in Figure 12 show the order of cycling at different stress amplitudes; i.e., firstly we cycled ice at higher stress
amplitudes (0.5-0.8 MPa), then at lower stress amplitudes (0.2-0.4 MPa). The results showed an increase in the hit
rate as stress amplitude increases, regardless of the sequence of cycling.

There are two possible sources of the noise detected. One is from microcracks while the other is from the
brine movement in pores during cycling analogous to a water-hammer mechanism. Since no remnant microcracks
were detected (Section 3.5), we suggest that generation of sound during cycling may be caused by the motion of brine;
i.e., when brine within pockets is forced to stop and to change the direction of motion during the change in the sense
of bending. Independence of hit rate from the sequence of cycling is consistent with this hypothesis.
**4. Discussion**
The results obtained from the experiments described in this paper show that the flexural strength of saline ice
can be increased upon reversed cyclic loading. Therefore, the same set of questions as for the freshwater ice should
be addressed here, i.e.: What governs the flexural strength of saline ice? Does crack propagation or crack nucleation
control the tensile strength? First of all, to understand the behavior of saline ice, it is important to recognize that
flexural strength in the present experiments is governed by the tensile strength, although greater by a factor of about
1.7 (Ashby and Jones, 2012). Secondly, the absence of remnant microcracks within the two parts of broken samples
(Section 3.5) indicates that crack nucleation controls the flexural strength, just as it appears to do for freshwater ice.
Indeed, this seems reasonable given the fact that freshwater ice comprises of ~95% by volume of the saline ice we
studied. Within the freshwater component, there is almost no solubility of salts (Weeks and Ackley, 1986). The
remainder of the saline ice is a mixture of air and brine. As was shown earlier, the microstructure of saline ice that we
grew is closely similar to the microstructure of sea ice. Pores lower the saline ice strength. However, the behavior of
S2 saline ice under cyclic loading is essentially the same as the behavior of S2 freshwater ice (Murdza et al., 2020c),
i.e. its strength increases at the same rate as freshwater ice upon cycling under a given amplitude of the outer fiber
stress. Hence, it is reasonable to assume that the strengthening mechanism for the saline ice is essentially the same as
that for the freshwater ice; i.e., due to the development of an internal back stress that originates from either dislocation
pileups or grain boundary sliding.

The maximum degree of strengthening in the case of saline ice is significantly lower than that for the
freshwater ice, although the slopes of the two data sets (rate of strength increase) in Figure 8 are nearly equivalent.
That difference may be explained by the structure of saline ice which limits maximum possible strengthening. In
contrast to the freshwater ice, saline ice has strongly irregular grains in shape and often elongated such that part of
one boundary may penetrate deeply inside another boundary and is surrounded by that boundary. As a result, grain
boundary sliding is significantly impeded, which may result in a smaller number of dislocations generated and
subsequent lower degree of strengthening, which is consistent with Iliescu et al. (2017) and Murdza et al. (2020c) who
stated that grain boundary sliding may lead to cyclic strengthening through the development of a back stress. Given



the significantly greater number of stress concentrators in saline ice, such as brine pockets and channels, saline ice is
more susceptible to premature failure, limiting the development of the back stress.

Flexural experiments conducted on saline ice of higher salinity (5.9±0.6 ppt) showed the importance of brine
features. Samples that were manufactured from the bottom of the ice puck were characterized by more frequent whitish
interconnected features (taken to be interconnected brine pockets) that often were the path for easy crack propagation.
Often samples were so weak that they failed before testing simply by handling. Interestingly, there were no
interconnected features in samples prepared from the top of an ice puck, which resulted in a difference of more than
a factor of three in strength between samples from top and bottom. Samples produced from saline ice of lower salinity
(3.0±0.9 ppt) also had whitish features; however, these features were spread more uniformly (on a macrosccoic scale)
across the sample, resulting in little difference in strength between the bottom and top samples.

It is worth noting again that a significantly greater fraction of saline ice samples failed in fatigue while pre-
conditioning compared with freshwater ice. This may be explained by the fact that freshwater ice was essentially free
from pores, brine pockets and other defects. Based on this observation, it appears that crack growth is not a significant
contribution to the fatigue life of saline ice under the conditions of our experiments..

Returning to the observations noted in the introduction, and to the results obtained from in situ cyclic loading
experiments on sea ice beams by (Bond and Langhorne, 1997; Haskell et al., 1996; Langhorne et al., 1998, 1999),
why did ice fail in the field under the wave action and cyclic loading, but strengthened upon cycling in our experiments
in the laboratory?

Although we do not know the process through which the ice sheet failed in the field, we expect that there are
many micro and macro cracks in natural sea ice. Indeed, thermally-induced tensile stresses can induce thermal
cracking in floating ice sheets (Evans and Untersteiner, 1971). Therefore, our sense is that the difference in ice
behavior under cyclic loading in situ in the field (Bond and Langhorne, 1997; Langhorne et al., 1998) and in the
laboratory in the present study is due to other types of defects than brine channels and pockets that are generated in
the field as a result of thermo-mechanical history of ice.
**5. Conclusions**
From new, systematic experiments on the flexural strength of sub-meter sized plates of S2 columnar-grained
saline ice stressed principally across the columns through reversed cyclic loading at a temperature of -10 °C and
frequencies in the range from 0.1 to 0.6 Hz, it is concluded that:
(i) The flexural strength of saline ice can be increased upon reversed cyclic loading by as much as 1.5 times.
(ii) The flexural strength of ice upon cycling scales linearly with the amplitude of the outer-fiber stress.
(iii) The fatigue life of saline ice is erratic and does not obey classical S-N behavior.



(iv)    Crack growth is not a significant contribution to the fatigue life of saline ice.
(v)     There is high variability in structure and strength through the thickness of a saline ice puck of higher salinity

(5.9±0.6 ppt).

(vi)    The strengthening mechanism for the saline ice is the same as for the freshwater ice.
(vii)   Acoustic emission hit rate during cycling at a constant stress amplitude is about constant.
(viii)   Acoustic emission hit rate during cycling increases with an increase of stress amplitude of cycling.

**Acknowledgements**

We acknowledge helpful discussions/communications with Prof. Harold Frost, Dr. Robert Gagnon, and
Dr. Daniel Iliescu. This work was supported by the US Department of the Interior-Bureau of Safety and Environmental
Enforcement (BSEE), contract no. E16PC00005 and by National Science Foundation (FAIN 1947-107).

**Author contributions:** AM, ES and CR designed the experiments and AM carried them out. AM prepared the manuscript with contributions from all co-authors.

**Competing interests:** The authors declare that they have no conflict of interest.

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

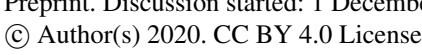



**Table 1. Physical properties of as-grown saline ice.**

| Material | Density [kg m$^{-3}$] | Average salinity [ppt] | Grain size [mm] |
|---|---|---|---|
| Saline ice (lower salinity) | 878±11 | 3.0±0.9 | 3.8±0.9 |
| Saline ice (higher salinity) | 897±10 | 5.9±0.6 | 3.6±1.1 |

**Table 2. Flexural strength of non-cycled saline ice at -10°C and a displacement rate of 0.1 mm/s.**

| Flex strength of ice of lower salinity (3.0±0.9 ppt) [MPa] | Depth [cm] | Flex strength of ice of higher salinity (5.9±0.6 ppt) [MPa] | Depth [cm] |
|---|---|---|---|
| 1.08 | — | 0.45 | 20 – 22.5 |
| 0.86 | — | 0.53 | 17.5 – 20 |
| 1.06 | — | 0.62 | 12.5 – 15 |
| 0.96 | — | 0.98 | 7.5 – 10 |
| 0.83 | 17 – 21 | 1.17 | 5 – 7.5 |
| 0.75 | 13.5 – 17 | 1.26 | 5 – 7.5 |
| 1.08 | 10 – 13.5 | 1.26 | 2.5 – 5 |
| 0.97 | 6.5 – 10 | 1.44 | 1 – 2.5 |
| 1.09 | 3 – 6.5 | 1.17 | — |
| | | | |
| Average | | Average | |
| 0.96±0.13 | | 0.98±0.36 | |



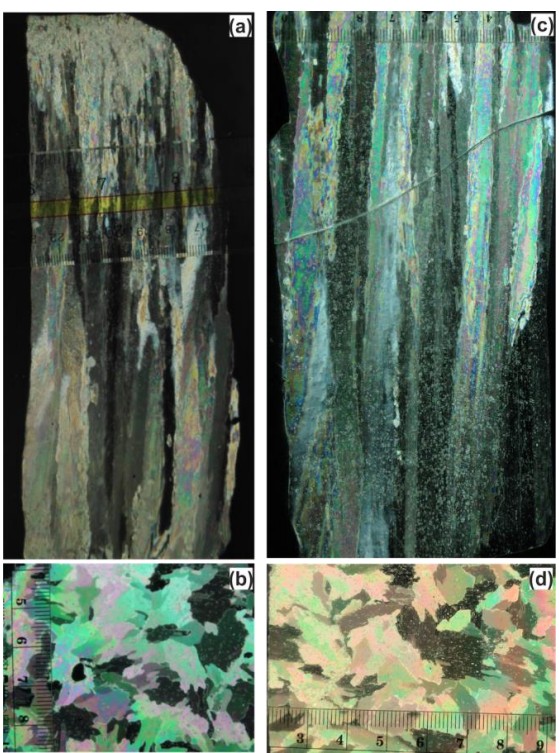

**Figure 1. Photographs of a vertically-oriented (a) and a horizontally-oriented (b) thin-sections (~1mm) of columnar-grained,**
**saline ice of lower salinity (3.0±0.9 ppt) as viewed between crossed-polarized filters; photographs of a vertically-oriented**
**(c) and a horizontally-oriented (d) thin-sections of saline ice of higher salinity (5.9±0.6 ppt).**

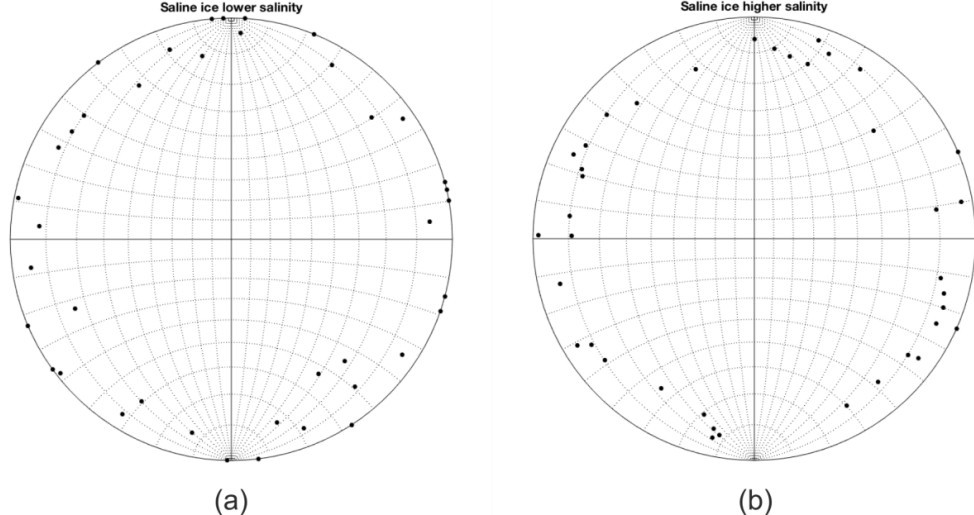

**Figure 2. Stereographic projection plots of crystal c-axis {0001} orientations in saline ice of lower (3.0±0.9 ppt) salinity (a)**
**and saline ice of higher (5.9±0.6 ppt) salinity (b).**



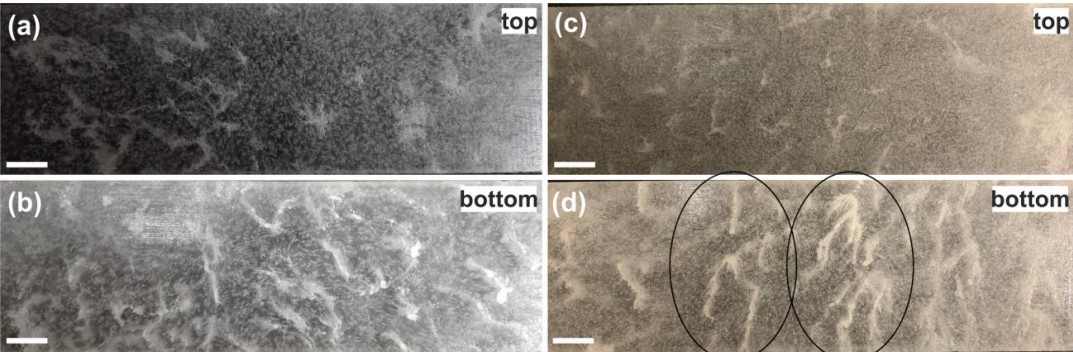

**Figure 3. Photographs of saline ice samples of lower salinity (3.0±0.9 ppt) from the top (a) and bottom (b) of an ice block**
**and saline ice samples of higher salinity (5.9±0.6 ppt) from the top (c) and bottom (d) of an ice block. The concentration of**
**whitish features along the width of a sample in (d) is shown inside circles which is a predominant place for a crack to initiate.**
**The columnar grains run in and out of the images. Scale bars: 20 mm.**

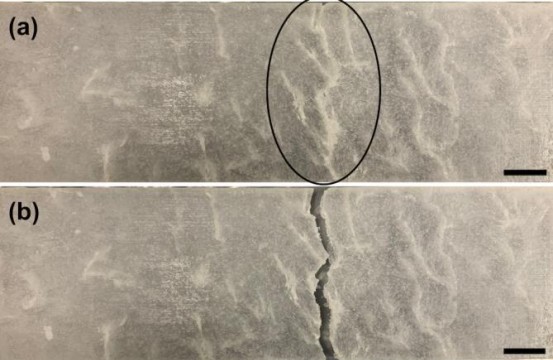

**Figure 4. Photograph of a sample from the bottom of an ice block of higher salinity (5.9±0.6 ppt) before cycling (a) and after**
**(b) failure. Note a crack that propagated along whitish features in the area in (a) depicted by the circle. Scale bars: 20 mm.**

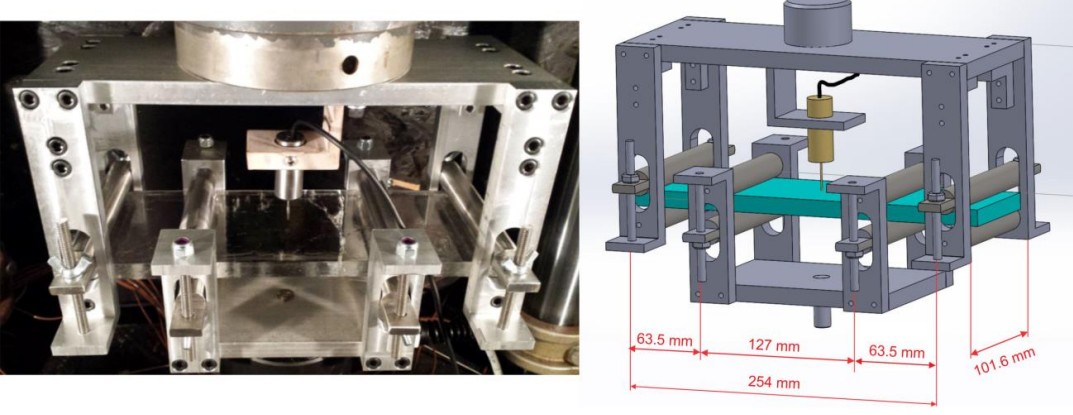

(a)                                                          (b)
**Figure 5. Photograph (a) and sketch (b) of the four-point bending apparatus connected to an MTS hydraulic testing system**
**(Iliescu et al., 2017; Murdza et al., 2020c). The upper part is attached to the frame of the machine while the mobile middle**
**part is attached through a fatigue-rated load cell to the piston. The apparatus is made from an aluminum alloy; the loading**
**cylinders are made from stainless steel.**





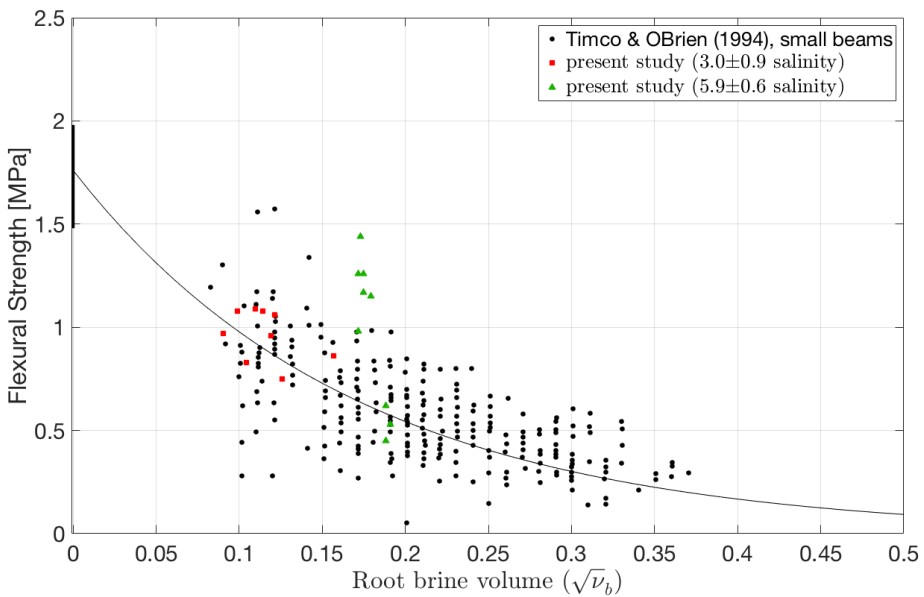

**Figure 6. Flexural strength of saline ice as a function of root brine volume for the ice grown in the present study and for**
**data from Timco and O'Brien (1994) for comparison.**

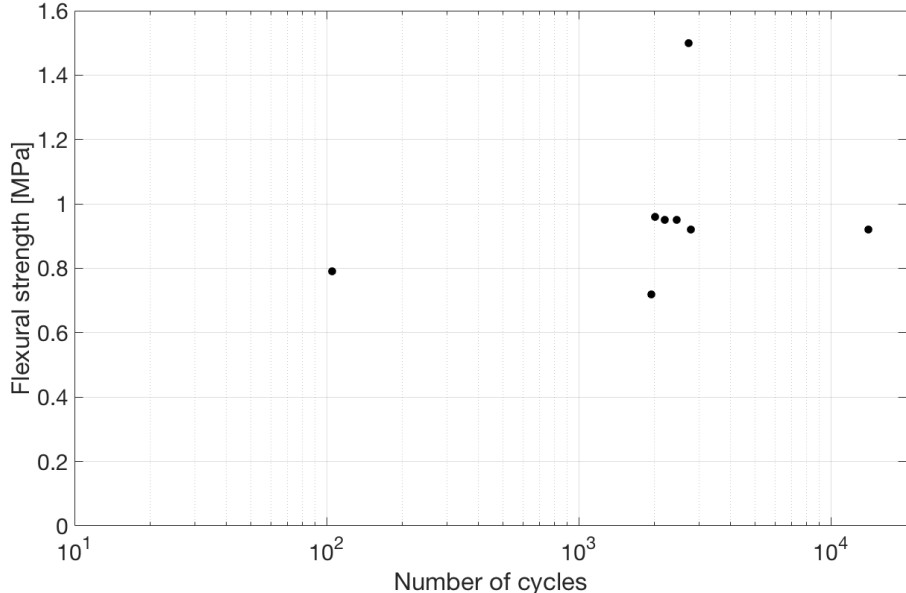

**Figure 7. Flexural strength and the corresponding number of cycles imposed for saline ice of lower salinity (3.0±0.9) ppt**
**cycled at 0.35 MPa outer-fiber stress amplitude at -10 ºC and 0.1 mm s⁻¹ outer-fiber center-point displacement rate.**



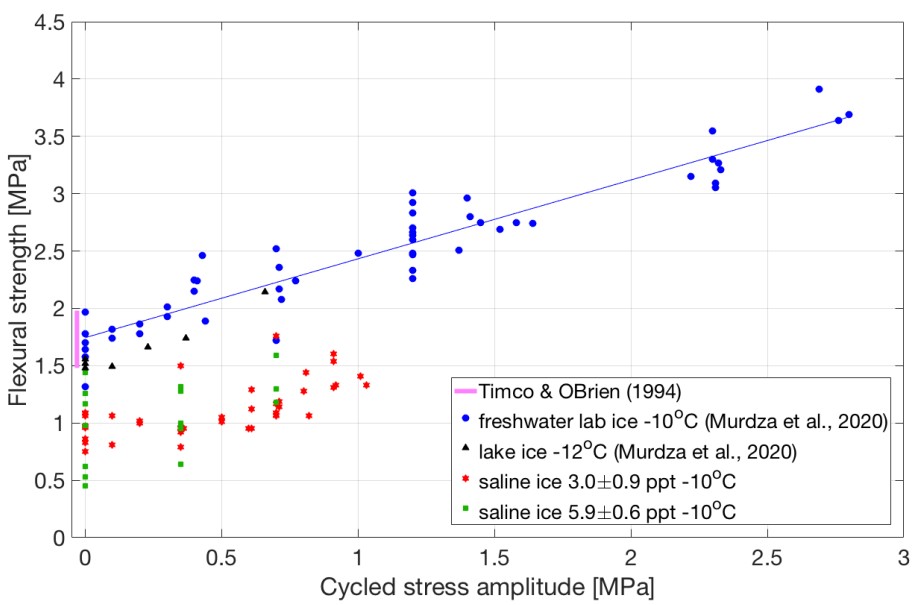

**Figure 8. Flexural strength of freshwater ice and saline ice of lower (3.0±0.9 ppt) and of higher (5.9±0.6 ppt) salinity as a function of reverse-cycled stress amplitude. Freshwater ice laboratory and lake data are taken from (Murdza et al., 2020c, 2020a). Red five-pointed stars and green squares represent tests performed on saline ice of lower and higher salinities, respectively, at 0.1 mm s⁻¹ and -10°C. During all depicted tests the ice did not fail during cycling and was broken by applying one unidirectional displacement until failure occurred.**

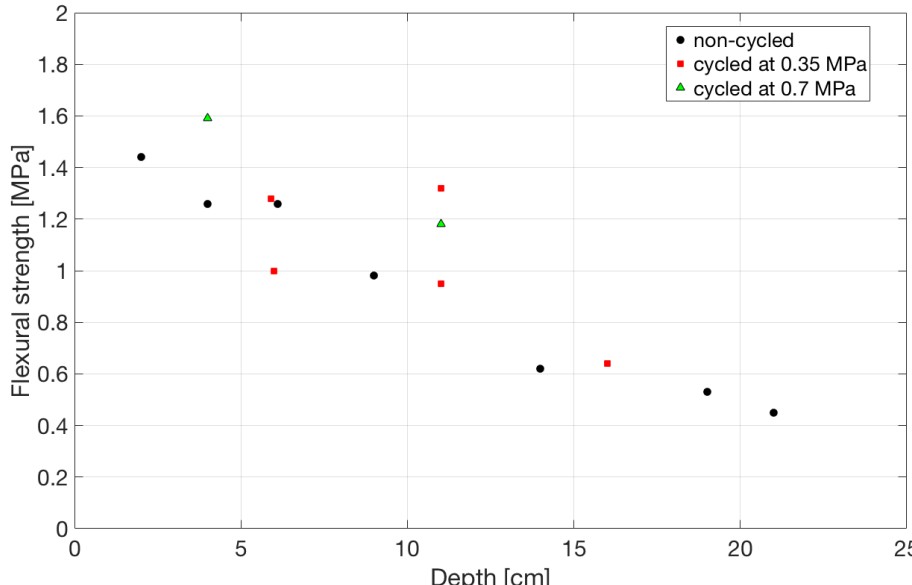

**Figure 9. Flexural strength as a function of position and fractional area of whitish features of saline ice samples of higher salinity (5.9±0.6 ppt) for different cyclic amplitudes.**



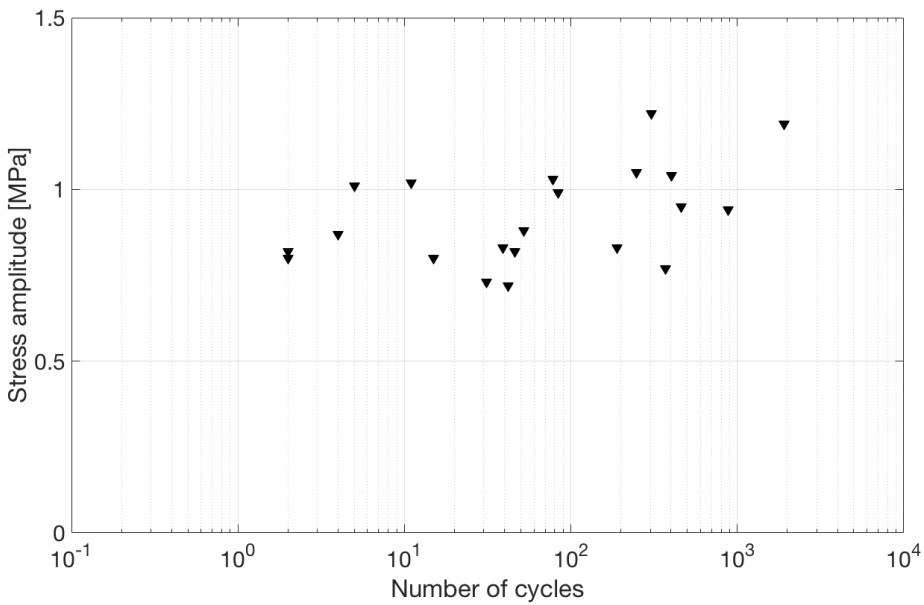

**Figure 10. Stress amplitude as a function of the number of cycles to fatigue fracture for saline ice of lower salinity (3.0±0.9 ppt) tested at -10ºC and 0.1 mm s⁻¹ outer-fiber center-point displacement rate.**

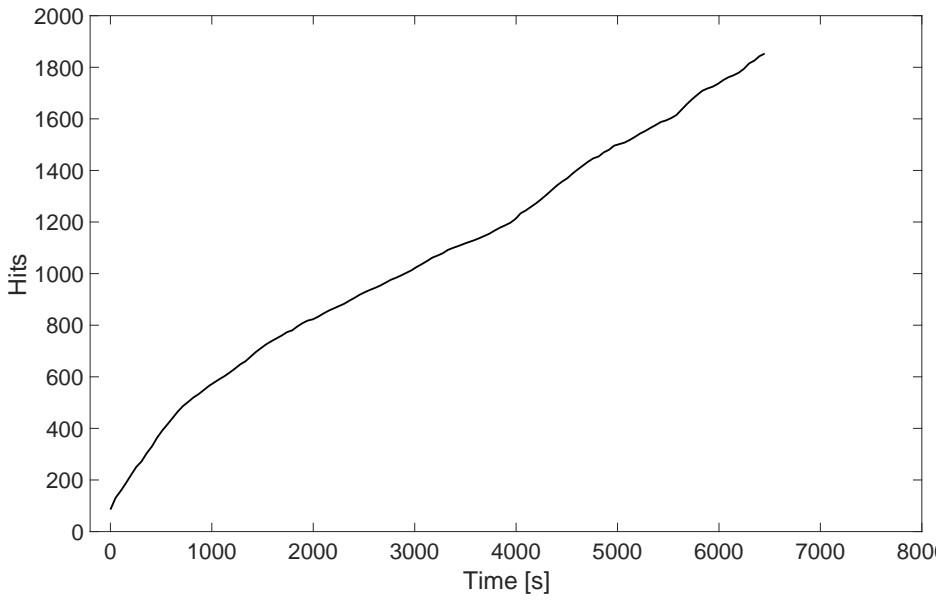

**Figure 11. Acoustic emissions (hits) against time for saline ice of lower salinity (3.0±0.9 ppt), cycled at a stress amplitude of 0.5 MPa at -10ºC at an outer-fiber displacement rate of 0.1 mm s⁻¹.**



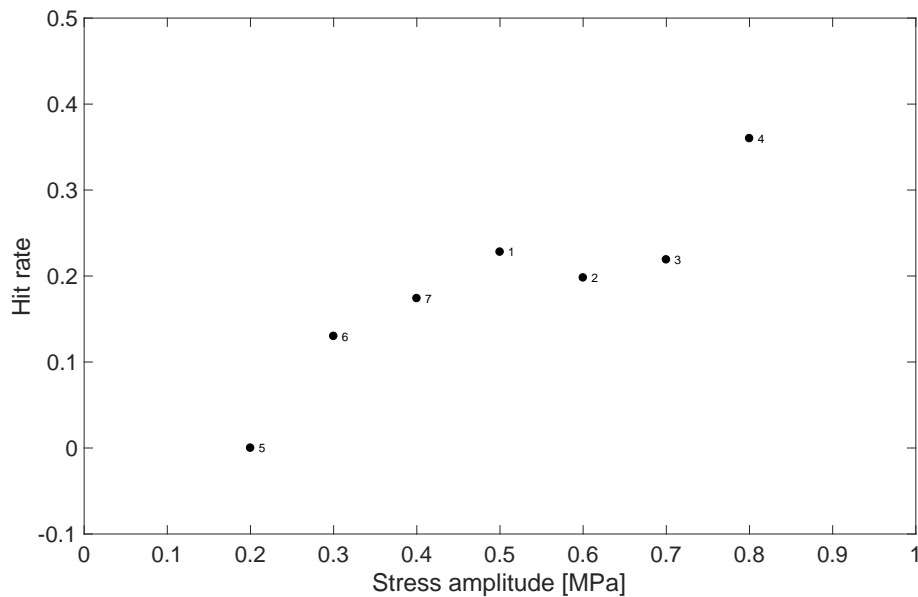

**Figure 12. Hit rate as a function of cycled stress amplitude for saline ice sample of lower salinity (3.0±0.9 ppt). Numbers**
**show the order of cycling at different stress amplitudes.**