# Peer review of "Behavior of Saline Ice under Cyclic Flexural Loading"

_The Cryosphere, 2020_

## Referee Comment (RC1) · Anonymous Referee #1 · 31 Dec 2020

Review comments on Behavior of Saline Ice under Cyclic Flexural Loading Andrii Murdza, Erland M. Schulson, Carl E. Renshaw tc-2020-300 31 Dec., 2020

General comments:

The experimental program is well described in the paper and the experiments themselves were well executed. The experiments uncovered an interesting and potentially important aspect of saline ice behavior and the authors are to be congratulated for developing this interesting line of work on the flexural behavior of ice. The graphics adequately portray the findings and the specimen images, particularly regarding the brine drainage features, are effective. The writing in general is clear and concise, with a few exceptions noted below. I believe that the paper presents valuable information and should ultimately go forward to publication after certain shortcomings are addressed

as detailed in the following.

Although I applaud the experimental effort, I have serious issues with certain assumptions and related conclusions that are put forth. The most significant issue is the discounting of microcracking as a viable damage mechanism in their test material despite the observation of prolific acoustic activity of the type that is generally associated with microcracking. Instead, a vaguely described AE mechanism due to liquid brine movement is put forth with no quantitative development. Additionally, it is concluded that the mechanism for cyclic-loading-induced strengthening is the same in both FW and saline ice without any valid proof beyond the rough similarity of the slope of the strengthening effect. In my view, these are fatal flaws in the manuscript as submitted. That being said, both relate to the interpretation of the findings and not to the actual results. Thus, it should be a relatively simple matter to address these issues in a revision. I strongly urge the authors to make the necessary changes.

Other improvements that would significantly strengthen the paper are: 1) include profiles of the physical properties of the high- and low-salinity ice sheets and specimen-specific salinities, 2) include information on the extreme fiber strain values at failure, and 3) include stress-strain data plots to illustrate basic constitutive behavior. The former measurements should be made as a matter of course and the constitutive information would serve the dual purpose of helping to relate the present effort to other work and the constitutive behavior would help inform the authors' understanding of potential underling mechanisms. I feel strongly that the above items are critical to a fuller understanding of the experimental results.

Mandatory changes:

Line 103: It would seem that the whitish features under discussion here are brine drainage features that consist of tubes that are typically filled with very fine-grained ice, rather than a collection of discrete brine inclusions as suggested in the text. Some clarification on this point is in order. These features constitute regions of weakness,

however, and the observed tendency for cracks to run through these regions is in line observations in the cited literature.

Lines 114-118: Since this paragraph addresses, albeit in a qualitative manner, the experimental results, it belongs in the results or discussion section.

Line 130: State the range of temperature variation about your -10C set point. Was the interior or surface temperature of any of the specimens monitored?

Lines 150-152: I find the use of "little evidence" to be unacceptably vague in this context. The two obvious quantities to check for changes in mechanical properties are the effective modulus over a load cycle and the hysteresis loop area. Were both of these quantities examined for the early and late stages of cyclic and were both found not to vary systematically?

Lines 163-166: This is an odd way to start the results section. I suggest that a comparison with other results be moved to the discussion section.

Line 177: Regarding the use of Eq.2, why use the 1967 expression for brine volume when it would be preferred to use the newer relationship presented in Cox and Weeks (1983)?

Line 186: Do you mean "larger brine volumes" here?

Lines 230-237: Although the analysis focuses on failure stress levels, the matter of extreme fiber strain at failure can provide critical information and should not be ignored. The observation that FW ice can be cyclically loaded beyond its quasistatic failure strength with much greater consistency than saline ice suggests that the cyclic loading conditions can more readily generate the necessary failure strain in saline ice than in FW ice. Consequently, I believe that it would be useful for the authors to put more thought on this matter. A few stress-strain plots of the early- and late-stage cyclic response are certainly called for here.

Line 253: Re the sentence beginning with "No systematic trend…" Drop the word "also"

and correct the verb situation. Possibly say "was plotted".

Line 269: Use "Microstructural" rather than "Experimental" in this section title.

Lines 284-295: The conclusion that microcracks did not occur in these experiments because they were not observed in the post-test thin sectioning is, in my view, incorrect and fatally detracts from the values of this paper. In my own experience on this topic, it is clear that microcracks occur extensively in saline and sea ice but they are not readily observed under a microscope because they immediately fill up with liquid brine upon formation. This results in a loss of contrast and renders them all but invisible. In addition, given the complex nature of the microstructure in saline ice, it may be difficult to distinguish between cracks and grown-in defects, and microfracture of the ligaments between brine inclusions can occur without much visual evidence. More to the point, the AE signal characteristics in saline and sea ice are virtually the same as observed in FW ice when microcracks occur. A useful technique in this context is to shine a light through the specimen during straining and take a video of the result. What you will see are flashes of light (reflections from the forming crack faces) that immediately disappear as brine is drawn into the crack. Not coincidentally, these events correspond to concurrently monitored AE activity. This exercise will also yield information on the general location of the microcracks. In light of the above considerations, I see virtually no support for the introduction of a new and novel AE source mechanism as described in the paper.

It would be beneficial for the authors to broaden their thinking about the tensile failure process in saline vs FW ice. The thinking in the paper seems to be predicated on there being some discrete and readily observable microcracking event that causes failure – as is typically seen in FW ice. In that case, tensile failures are generally controlled by crack nucleation with only minor precursor events. Alternatively, the prolific flaw structure in saline ice allows for damage accumulation due to a great number of microcracks in regions of microstructural weakness. The brine drainage features in the test specimens constitute regions of high porosity and thus provide favorable sites

for the concentration of such damage. Failure occurs when one of these sites can no longer support the applied stress and a macrocrack emerges from the damage zone & propagates. Strain data from these experiments should shed light on this matter – especially in comparison with their FW ice data, which presumably show considerably less straining that the saline ice experiments.

Line 333-337: The similarity in the slope of the cyclic loading effect between FW and saline ice does not support the conclusion that both are the result of the same mechanism. The vast differences in the microstructures of the two ice types militates against such a simple approach. What I suspect is much more likely is that cyclic loading, when conducted at appropriate stress levels, acts to blunt the effectiveness of stress concentrations within both materials, but by dissimilar mechanisms. Some of the FW data with which I am familiar show clear evidence of a process known as dislocation punch-out, in which dislocations are produced in large numbers from grain boundary triple points and ledges during a load cycle. They act to relieve the local stress level sufficiently to prevent crack nucleation. In some cases, the dislocations thus produced can collapse back to their sources upon unloading. It would be worth examining your FW ice data for evidence of this process. On the other hand, due to the inherent weakness of the saline ice microstructure, the same sort of microstructural stress relief most likely occurs through localized damage via microcracking. In this way, two completely difference processes can ultimately have similar effects on the flexural strength.

Line 344: Please become acquainted with the literature on grain boundary sliding in various ice types. For polycrystalline ice (FW, saline or naturally occurring sea ice) of grain sizes over about 2 mm, the extent of grain boundary sliding is remarkably consistent. Consequently, it is difficult to invoke variations in that process in the argument being made here.

Line 361-362: This statement may be reasonable for small-scale laboratory experiments, but I had personally witnessed incremental crack growth under cyclic loading in full-scale field experiments on sea ice. Thus, it would be prudent to qualify this

statement as applicable to the scale of your experiments.

Line 380: Suggest changing "upon" to "subsequent to".

Line 385: This conclusion is based on weak assumptions and cannot logically be drawn from the experimental observations. Objectively, the best one can say given the lack of definitive proof of the underlying failure mechanism in saline ice, is that the increase in flexural strength of FW and saline ice attributable to pre-failure load cycling is roughly equivalent. With regard to figures:

Figure 11: The y-axis label should be "Cumulative hits" if that's the quantity being plotted. Figure 12: The y-axis label needs units.

Suggested changes:

Line 19: This first sentence needs a citation or two. One or two of the standard texts on fatigue & failure of materials in general should suffice.

Line 90: For practical considerations, it is understandable that the skeletal layer was discarded from the specimens, but the resulting material will not be representative of in-situ conditions viz-a-viz the flaw structure. A comment on the potential effects of removing this material would thus be appropriate.

Line 92: Regarding Figure 1 and the grain sizes reported in Table 1: It is clear from the vertical thin sections that the grain size increases significantly with depth, as is usually the case in congelation ice. Consequently, the reported grain sizes should be associated with a specific depth in the ice sheet. Moreover, since the specimens came from a range of depths in the parent ice sheets, grain size likely varied considerably across the specimen population & values should be reported.

Lines 121-123: In my view, storing the ice blocks on their sides is not going to inhibit brine drainage to any appreciable extent. Until the ice is cold enough to close off brine drainage pathways, brine will drain out downward or sideways and storing it in other than its growth orientation runs the risk of establishing brine inclusions with orientations

that do not occur naturally. The general practice is to store specimens in their growth orientation and keep them as cold as possible. On this point, it will be important to state how long the specimens were in storage.

Lines 139-143: It would be helpful in this paragraph to include the peak values of the extreme fiber stress and strain associated with the range in applied stress rates. Additionally, the description indicates that the test machine was operated in either displacement control (of the loading piston) or strain control (from a specimen-mounted transducer). Please specify which method was used.

Lines 154-161: For the Type II tests, was the stress level increased on the fly or was the loading stopped while the settings were changed?

Lines 225-228: Note that the approximate threshold of 0.4 MPa for the cyclic loading effects agrees well with observations of the onset of significant AE activity under cyclic loading of sea ice cores presented in the Cole and Dempsey (2006) paper that is cited.

---

## Referee Comment (RC2) · Anonymous Referee #2 · 18 Jan 2021

General Comments: The key contribution of the paper, in addition to provide new experimental data on flexural strength of saline ice under cyclic loading, is discussing fatigue and its apparent non-classical manifestation under the cyclic loading conditions of these experiments. The key finding was that cyclic flexural stressing of a saline ice beam leads to an observed increase of flexural strength. The manuscript merits publication, but more description and explanation of the tests is required first. Specific items are identified here and in the "Line" items following. The authors present a comprehensive literature review of cyclic loading in the context of the breakup of ice sheets under ocean swell. Most of the literature on sea ice has been on weakening under cyclic loading, interpreted in terms of an S-N curve and an endurance limit (cyclic stress limiting value under which failure would not occur). Fully understanding the experiments

and analysis in this paper requires familiarity with the authors' previous publications on this strengthening phenomenon in freshwater ice, including several which were just published in 2020. The paper requires a more detailed description and explanation of the tests and results. Ice, being a high temperature material with relatively large grains, I would expect time and strain are critical parameters in characterizing its behaviour. Your loading periods are from 1 to 10 s, certainly providing time for delayed-elastic and plastic strains. A representative plot of force and deflection versus time should be added to show the reader whether time and strain are significant, or can be ignored. Figure 7 shows that for the low salinity ice and Type 1 cycling at 0.35 MPa no strengthening was observed. Provide an additional figure where results show a clear example of significant strengthening for saline ice. From Figure 8 it appears a stress amplitude of 0.7 MPa or higher is required to see a strengthening beyond 0.96 MPa simple flexural strength. This means that for low salinity ice you had to go to Type 2 cycling to get strengthening. The results in Figure 8 are hard to follow, with the results of many different tests jumbled together. Similarly, Figure 9 mixes different tests without saying how many cycles were conducted before loading to failure. Add a table which provides the test results as a function of the cycle type (Type 1 or 2, and the actual program of the Type 2 cycling for that test, number of cycles, frequency, time). This would greatly improve the paper. The purpose of 4-point loading is to create a centre section with a constant bending moment. Did the failures occur at random locations between the two inner loading cylinders? Provide some observations on failure location. More of an aside, the authors may be interested in an observation in the book by D. Masterson published in 2019, "The Story of Offshore Arctic Engineering". It mentions experience in the field of moving a lightly loaded vehicle back and forth on a floating ice road before moving a greater load along it, as a means of improving the load bearing capacity of the ice road. Your work on cyclic flexural loading seems to provide an explanation for this field experience.

Specific Comments: Line 21; suggest adding 'and failure' to the end of the sentence. Line 35; the sentence starting with 'For instance...' is not clear, are you saying that

the structure is being fatigued, or the ice? Line 70; you mention 'recovery', what is being recovered and does that mean increasing or decreasing strength? Line 78; Your experiments were performed on and analysed as beams, change 'plates' to beams. Lines 80-90; You mention that ice plates were grown in a circular tank, how deep was the tank? Did you seed the sheet? You mention melt water salinities, were the values given from the tested beams or samples from the whole ice sheet? Density of the ice beams should also be provided, that would help distinguish between brine pockets and air filled voids. Line 121; Make clear the orientation of these blocks in the original sheet, presumably the long dimension was in plane of the plate. Line 132, Figure 5; This figure indicates that deflection at the centre-point of the beam was measured with respect to the outer pair of loading cylinders of the four-point loading apparatus, this introduces an error. You should be measuring the deflection of the beam with respect to the inner pair of loading cylinders. Line 151; 'softening', what do you mean? Rewrite sentenced to be clearer. Line 168; were the test beams always in the same orientation as in the original puck? Also for the simple flexural tests, or the final loading to failure after Type 1 or 2 cycling, was the top or bottom surface the one in tension? Line 186; sp. 'contain' Line 289; would brine not fill cracks making them difficult, if not impossible, to detect visually; also if I understand the orientation of the thin section, the chance of having a crack in it would be rare. Line 300; could emissions also originate from grain boundary movements? Line 317; 'water hammer' is usually associated with pressure waves in a fluid in a closed system, If you are proposing brine movements in pores, some further explanation of the mechanism is needed. Line 331; where is the air and brine distributed, separate pockets or both in the same pocket? Line 347; explain how a brine pocket or channel makes saline ice more susceptible to premature failure. Brine pockets are very rounded, have a much larger radius and lower stress concentration than for a crack. Line 352; this discussion of brine pockets is very subjective, careful thin sectioning could have provided more definitive information on the grain structure. Line 376; the tests were done on beams, don't refer to them as plates. Line 380; does this conclusion apply to all ice, fresh water and saline? Line 581, Fig. 12; why

a negative value of hit rate, start the ordinate at 0.0, also state the units for hit rate on this axis.

---

## Author Comment (AC1) · 7 Mar 2021

**Responses to comments by reviewer of manuscript tc-2020-300 "Behavior of Saline Ice under Cyclic Flexural Loading"**

We sincerely thank anonymous referee for valuable comments/suggestions on our work. The comments are constructive and insightful. We have modified our manuscript according to them. Please, see all the responses in red.

**Comments from Referee # 1**

**General comments:**

The experimental program is well described in the paper and the experiments themselves were well executed. The experiments uncovered an interesting and potentially important aspect of saline ice behavior and the authors are to be congratulated for developing this interesting line of work on the flexural behavior of ice. The graphics adequately portray the findings and the specimen images, particularly regarding the brine drainage features, are effective. The writing in general is clear and concise, with a few exceptions noted below. I believe that the paper presents valuable information and should ultimately go forward to publication after certain shortcomings are addressed as detailed in the following.

Although I applaud the experimental effort, I have serious issues with certain assumptions and related conclusions that are put forth. The most significant issue is the discounting of microcracking as a viable damage mechanism in their test material despite the observation of prolific acoustic activity of the type that is generally associated with microcracking. Instead, a vaguely described AE mechanism due to liquid brine movement is put forth with no quantitative development. Additionally, it is concluded that the mechanism for cyclic-loading-induced strengthening is the same in both FW and saline ice without any valid proof beyond the rough similarity of the slope of the strengthening effect. In my view, these are fatal flaws in the manuscript as submitted. That being said, both relate to the interpretation of the findings and not to the actual results. Thus, it should be a relatively simple matter to address these issues in a revision. I strongly urge the authors to make the necessary changes.

Other improvements that would significantly strengthen the paper are: 1) include profiles of the physical properties of the high- and low-salinity ice sheets and specimen specific salinities, 2) include information on the extreme fiber strain values at failure, and 3) include stress-strain data plots to illustrate basic constitutive behavior. The former measurements should be made as a matter of course and the constitutive information would serve the dual purpose of helping to relate the present effort to other work and the constitutive behavior would help inform the authors' understanding of potential underling mechanisms. I feel strongly that the above items are critical to a fuller understanding of the experimental results.

**Mandatory changes:**

1. Line 103: It would seem that the whitish features under discussion here are brine drainage features that consist of tubes that are typically filled with very fine-grained ice, rather than a collection of discrete brine inclusions as suggested in the text. Some clarification on this point is in order. These features constitute regions of weakness, however, and the observed tendency for cracks to run through these regions is in line observations in the cited literature.

We changed this sentence and added that interconnected brine pockets can be filled with very fine-grained ice as suggested.

2. Lines 114-118: Since this paragraph addresses, albeit in a qualitative manner, the experimental results, it belongs in the results or discussion section.

We moved these sentences to the Results section.

3. Line 130: State the range of temperature variation about your -10C set point. Was the interior or surface temperature of any of the specimens monitored?

We added ±0.5°C variation in cold room temperature. We also monitored the surface temperature of a few specimens before and during cycling with an infrared digital thermometer whose readings were also within ±0.5C.

4. Lines 150-152: I find the use of "little evidence" to be unacceptably vague in this context. The two obvious quantities to check for changes in mechanical properties are the effective modulus over a load cycle and the hysteresis loop area. Were both of these quantities examined for the early and late stages of cyclic and were both found not to vary systematically?

We changed this sentence in the following way: "*Figure 6 shows measurements of load and of displacement versus time at the beginning and near the end of cycling before specimen failure of a lower-salinity specimen (3.0±0.9 ppt). The measurements detected no softening. (According to Bažant et al. (1984) softening is a decline of stress at increasing strain or, in our case, an increase of strain during cycling at constant stress amplitude during the tests). The absence of detectable softening during cycling of the saline ice is reminiscent of the absence of softening during the cycling of freshwater ice (Iliescu et al., 2017; Murdza et al., 2020b).*". We also added a load-time and displacement-time plots, new Figure 6, at the beginning and at the end of cycling (where end of cycling in this figure corresponds to premature fatigue failure). This result was reproducible and obtained systematically. The area of the hysteresis loop did not change over the time of cycling.

[Figure]

5. Lines 163-166: This is an odd way to start the results section. I suggest that a comparison with other results be moved to the discussion section.

We agree and moved this part to the discussion section.

6. Line 177: Regarding the use of Eq.2, why use the 1967 expression for brine volume when it would be preferred to use the newer relationship presented in Cox and Weeks (1983)?

We compared values of brine volume fraction obtained through Eq2 in the manuscript and using expression provided in Cox and Weeks (1983) as suggested for a few specimens. The results obtained were similar and not significantly different. Therefore, we think that both methods can be used. The reason why we used an expression by Frankenstein and Garner (1967) is because it does not require measurements of density. Unfortunately, we did not measure density for some of our specimens; in addition, we believe that density measurements may introduce an extra error in brine fraction calculations.

7. Line 186: Do you mean "larger brine volumes" here?

By larger volumes we meant larger specimen size. We corrected the text accordingly.

8. Lines 230-237: Although the analysis focuses on failure stress levels, the matter of extreme fiber strain at failure can provide critical information and should not be ignored. The observation that FW ice can be cyclically loaded beyond its quasistatic failure strength with much greater consistency than saline ice suggests that the cyclic loading conditions can more readily generate the necessary failure strain in saline ice than in FW ice. Consequently, I believe that it would be

useful for the authors to put more thought on this matter. A few stress-strain plots of the early- and late-stage cyclic response are certainly called for here.

As noted in comment 4 above, we added load-time and displacement-time plots of the early- and late-stage (right before failure) cyclic response as suggested above (since the specimen dimensions were not changing during cycling, the load and displacement are equivalent to stress and strain). The results revealed that strain amplitude did not increase over time; moreover, during the last cycle before premature fatigue failure the strain amplitude also remained the same as at the beginning of cycling.

9.Line 253: Re the sentence beginning with "No systematic trend. . ." Drop the word "also" and correct the verb situation. Possibly say "was plotted".

We corrected the sentence by removing the word "also" and replaced "plotted" with "was plotted".

10.Line 269: Use "Microstructural" rather than "Experimental" in this section title.

We have used the term "Microstructural" instead of "Experimental".

11. Lines 284-295: The conclusion that microcracks did not occur in these experiments because they were not observed in the post-test thin sectioning is, in my view, incorrect and fatally detracts from the values of this paper. In my own experience on this topic, it is clear that microcracks occur extensively in saline and sea ice but they are not readily observed under a microscope because they immediately fill up with liquid brine upon formation. This results in a loss of contrast and renders them all but invisible. In addition, given the complex nature of the microstructure in saline ice, it may be difficult to distinguish between cracks and grown-in defects, and microfracture of the ligaments between brine inclusions can occur without much visual evidence. More to the point, the AE signal characteristics in saline and sea ice are virtually the same as observed in FW ice when microcracks occur. A useful technique in this context is to shine a light through the specimen during straining and take a video of the result. What you will see are flashes of light (reflections from the forming crack faces) that immediately disappear as brine is drawn into the crack. Not coincidentally, these events correspond to concurrently monitored AE activity. This exercise will also yield information on the general location of the microcracks. In light of the above considerations, I see virtually no support for the introduction of a new and novel AE source mechanism as described in the paper.

It would be beneficial for the authors to broaden their thinking about the tensile failure process in saline vs FW ice. The thinking in the paper seems to be predicated on there being some discrete and readily observable microcracking event that causes failure – as is typically seen in FW ice. In that case, tensile failures are generally controlled by crack nucleation with only minor precursor events. Alternatively, the prolific flaw structure in saline ice allows for damage accumulation due to a great number of microcracks in regions of microstructural weakness. The brine drainage features in the test specimens constitute regions of high porosity and thus provide favorable sites for the concentration of such damage. Failure occurs when one of these sites can no longer support the applied stress and a macrocrack emerges from the damage zone &

propagates. Strain data from these experiments should shed light on this matter – especially in comparison with their FW ice data, which presumably show considerably less straining that the saline ice experiments.

In view of the reviewer's suggestions, we modified Section 3.6 Acoustic Emissions in the following way:

There are four possible sources of the noise detected. One is from microcracking. We imagine that microcracks form in regions of mechanical weakness which results in accumulation of damage that we detected via the AE method. Specifically, the brine drainage whitish features discussed above in the test specimens constitute regions of high porosity and thus provide favorable sites for the concentration of such damage. Failure may occur when one of these sites can no longer support the applied stress and a microcrack emerges from the damage zone and propagates. It is possible that newly formed microcracks are stable until a critical length is reached (Cannon et al., 1990; Schulson et al., 1991), at which point the crack growth ensues. The reason that microcracks were not observed under the optical microscope may be because they filled up with liquid brine upon formation which results in a loss of contrast. A second possible explanation for the acoustic emissions is the motion and friction of very fine particles of ice which may have been entrapped inside brine drainage features, as mentioned above. A third possibility is microcracking along grain boundaries due to grain boundary sliding (Elvin and Shyam Sunder, 1996; Goldsby and Kohlstedt, 1997; Mulmule and Dempsey, 1997; Schulson et al., 1997; Weiss and Schulson, 2000). The fourth possible explanation—consistent with the non-history dependence of the hit rate (new Figure 13) - is a kind of water-hammer effect in which brine entrapped within pockets impacts the wall, first in one direction and then another. None of these possibilities can be evaluated based upon the limits of the present observations. We refrain, therefore, from further speculation on this point.

12.Line 333-337: The similarity in the slope of the cyclic loading effect between FW and saline ice does not support the conclusion that both are the result of the same mechanism. The vast differences in the microstructures of the two ice types militates against such a simple approach. What I suspect is much more likely is that cyclic loading, when conducted at appropriate stress levels, acts to blunt the effectiveness of stress concentrations within both materials, but by dissimilar mechanisms. Some of the FW data with which I am familiar show clear evidence of a process known as dislocation punch-out, in which dislocations are produced in large numbers from grain boundary triple points and ledges during a load cycle. They act to relieve the local stress level sufficiently to prevent crack nucleation. In some cases, the dislocations thus produced can collapse back to their sources upon unloading. It would be worth examining your FW ice data for evidence of this process. On the other hand, due to the inherent weakness of the saline ice microstructure, the same sort of microstructural stress relief most likely occurs through localized damage via microcracking. In this way, two completely difference processes can ultimately have similar effects on the flexural strength.

Regarding the suggested strengthening mechanism via localized damage, we added the following sentences (lines 333-346): *"However, we should point out that there is a possibility for a different strengthening mechanism. Due to the inherent weakness of the saline ice microstructure, the microstructural stress relief may occur through localized damage via microcracking mentioned above. More research, however, is needed to examine this hypothesis."*. We agree that this mechanism (hypothesis) is possible, although we do not have any strong evidence/observations that support this argument as the correct mechanism. We still think, however, that the fact that the strengthening slopes for both fresh and saline ice are similar hints that the strengthening mechanism may be similar in the two materials.

13. Line 344: Please become acquainted with the literature on grain boundary sliding in various ice types. For polycrystalline ice (FW, saline or naturally occurring sea ice) of grain sizes over about 2 mm, the extent of grain boundary sliding is remarkably consistent. Consequently, it is difficult to invoke variations in that process in the argument being made here.

We reviewed the literature and, therefore, removed this argument from the manuscript.

14. Line 361-362: This statement may be reasonable for small-scale laboratory experiments, but I had personally witnessed incremental crack growth under cyclic loading in full-scale field experiments on sea ice. Thus, it would be prudent to qualify this statement as applicable to the scale of your experiments.

At the end of this sentence we added *"under the conditions of our experiments"*.

15. Line 380: Suggest changing "upon" to "subsequent to".

We corrected the wording as suggested.

16. Line 385: This conclusion is based on weak assumptions and cannot logically be drawn from the experimental observations. Objectively, the best one can say given the lack of definitive proof of the underlying failure mechanism in saline ice, is that the increase in flexural strength of FW and saline ice attributable to pre-failure load cycling is roughly equivalent.

We modified this conclusion as suggested.

17. With regard to figures:

Figure 11: The y-axis label should be "Cumulative hits" if that's the quantity being plotted.
Figure 12: The y-axis label needs units.

We made changes respectively.

**Suggested changes:**

18.Line 19: This first sentence needs a citation or two. One or two of the standard texts on fatigue & failure of materials in general should suffice.
We added a few citations.

19. Line 90: For practical considerations, it is understandable that the skeletal layer was discarded from the specimens, but the resulting material will not be representative of in-situ conditions viz-a-viz the flaw structure. A comment on the potential effects of removing this material would thus be appropriate.

We added the following clarification: *"For practical considerations, the bottom, skeletal layer of ice of about 7-10 cm was discarded as it was slushy and weak; we also believe that the skeletal layer in nature does not play a significant role in supporting the load during loading"*.

20. Line 92: Regarding Figure 1 and the grain sizes reported in Table 1: It is clear from the vertical thin sections that the grain size increases significantly with depth, as is usually the case in congelation ice. Consequently, the reported grain sizes should be associated with a specific depth in the ice sheet. Moreover, since the specimens came from a range of depths in the parent ice sheets, grain size likely varied considerably across the specimen population & values should be reported.

We mentioned in the text (lines 108-109) the grain size variation along the depth of ice pucks. We also added to the text (lines 99-100) that the top layer with small grain size was not used for the specimen preparation "because it was seeded and its grain size was considerably smaller and its microstructure thus different from the rest of the ice puck".

21.Lines 121-123: In my view, storing the ice blocks on their sides is not going to inhibit brine drainage to any appreciable extent. Until the ice is cold enough to close off brine drainage pathways, brine will drain out downward or sideways and storing it in other than its growth orientation runs the risk of establishing brine inclusions with orientations that do not occur naturally. The general practice is to store specimens in their growth orientation and keep them as cold as possible. On this point, it will be important to state how long the specimens were in storage.

We added that we stored ice for time periods ranging from 1 to 10 weeks. We will consider this suggestion for the future experiments.

22.Lines 139-143: It would be helpful in this paragraph to include the peak values of the extreme fiber stress and strain associated with the range in applied stress rates. Additionally, the description indicates that the test machine was operated in either displacement control (of the loading piston) or strain control (from a specimen-mounted transducer). Please specify which method was used.

We added (lines 149-150) peak values of the extreme fiber stress and strain associated with the range in applied stress rates. Regarding displacement vs strain control, it says in the text (line 137) that constant displacement rate was used, i.e. displacement rate of an actuator was constant during cycling.

23.Lines 154-161: For the Type II tests, was the stress level increased on the fly or was the loading stopped while the settings were changed?

The loading was stopped for ~15 sec to change settings. This point is noted in the revised manuscript.

24.Lines 225-228: Note that the approximate threshold of 0.4 MPa for the cyclic loading effects agrees well with observations of the onset of significant AE activity under cyclic loading of sea ice cores presented in the Cole and Dempsey (2006) paper that is cited.

We added the following sentence: *"Interestingly, this apparent threshold is similar in magnitude to the stress that marks the onset of significant AE activity under cyclic loading of sea ice cores (Cole and Dempsey, 2006)".*

---

## Author Comment (AC2) · 7 Mar 2021

**Responses to comments by reviewer of manuscript tc-2020-300 "Behavior of Saline Ice under Cyclic Flexural Loading"**

We sincerely thank anonymous referee for valuable comments/suggestions on our work. The comments are constructive and insightful. We have modified our manuscript according to them. Please, see all the responses in red.

**Comments from Referee # 2**

General Comments:

The key contribution of the paper, in addition to provide new experimental data on flexural strength of saline ice under cyclic loading, is discussing fatigue and its apparent non-classical manifestation under the cyclic loading conditions of these experiments. The key finding was that cyclic flexural stressing of a saline ice beam leads to an observed increase of flexural strength. The manuscript merits publication, but more description and explanation of the tests is required first. Specific items are identified here and in the "Line" items following. The authors present a comprehensive literature review of cyclic loading in the context of the breakup of ice sheets under ocean swell. Most of the literature on sea ice has been on weakening under cyclic loading, interpreted in terms of an S-N curve and an endurance limit (cyclic stress limiting value under which failure would not occur). Fully understanding the experiments and analysis in this paper requires familiarity with the authors' previous publications on this strengthening phenomenon in freshwater ice, including several which were just published in 2020. The paper requires a more detailed description and explanation of the tests and results. Ice, being a high temperature material with relatively large grains, I would expect time and strain are critical parameters in characterizing its behaviour. Your loading periods are from 1 to 10 s, certainly providing time for delayed-elastic and plastic strains. A representative plot of force and deflection versus time should be added to show the reader whether time and strain are significant, or can be ignored.

We added plots of force and deflection vs time for short time periods at the beginning of cycling and near the end of cycling before failure. In the text, we added the following paragraph (lines 171-176): "Figure 6 shows measurements of load and of displacement versus time at the beginning and near the end of cycling before specimen failure of a lower-salinity specimen (3.0±0.9 ppt). The measurements detected no softening. (According to Bažant et al. (1984) softening is a decline of stress at increasing strain or, in our case, an increase of strain during cycling at constant stress amplitude during the tests). The absence of detectable softening during cycling of the saline ice is reminiscent of the absence of softening during the cycling of freshwater ice (Iliescu et al., 2017; Murdza et al., 2020b)".

Figure 7 shows that for the low salinity ice and Type 1 cycling at 0.35 MPa no strengthening was observed. Provide an additional figure where results show a clear example of significant strengthening for saline ice.

The goal of this figure is to show that the number of cycles imposed (once above a certain threshold in the number of cycles) does not affect ice flexural strength, not to provide any information on strengthening. We chose the stress amplitude of 0.35MPa for this purpose for simplicity as none of the specimens failed at such a low stress amplitude. The results are in line with the results obtained earlier on freshwater ice; it makes sense to conclude that beyond a certain number of cycles the actual number of cycles does not affect flexural strength at higher stress amplitudes as well. Therefore, we do not think that an additional figure, similar to Figure 7 but with a greater cycling stress amplitude, is needed. Figure 8, on the other hand, shows the strengthening of ice upon cycling. It is clear from Figure 8 that at higher stress amplitudes ice is stronger after cycling when compared with non-cycled ice.

From Figure 8 it appears a stress amplitude of 0.7 MPa or higher is required to see a strengthening beyond 0.96 MPa simple flexural strength. This means that for low salinity ice you had to go to Type 2 cycling to get strengthening. The results in Figure 8 are hard to follow, with the results of many different tests jumbled together.

Yes, this is correct that a stress amplitude of 0.7 MPa or higher is required to detect a **significant** strengthening effect. The reason why we included freshwater results in Figure 8 is to compare the ice behavior and to point out similarities for freshwater ice and saline ice of both salinities.

Similarly, Figure 9 mixes different tests without saying how many cycles were conducted before loading to failure. Add a table which provides the test results as a function of the cycle type (Type 1 or 2, and the actual program of the Type 2 cycling for that test, number of cycles, frequency, time). This would greatly improve the paper.

We added to the figure caption of Figure 9 (new Figure 10) information on how many cycles were imposed to obtain the data shown in the figure .Regarding the second part of the comment, we added the following statement (lines 163-167) where we clarify how specimens were cycled during Type II loading: *"To cycle ice samples at stress amplitudes above 0.7 MPa, we first pre-conditioned them through step-loading Type II procedure at progressively higher stress amplitude levels, i.e. we cycled specimens for ~300 times at each of the following stress amplitudes: 0.7, 0.75, 0.8, 0.85 MPa and so on either until failure occurred or until a specific value of stress amplitude set by the operator".* Frequency and time for each specimen depends slightly on its dimensions since we keep displacement rate constant (not frequency).

The purpose of 4-point loading is to create a centre section with a constant bending moment. Did the failures occur at random locations between the two inner loading cylinders? Provide some observations on failure location.

We added the following sentence (lines 181-185): *"Failure generally occurred at random locations between the two inner loading cylinders and rarely either below or slightly outside the loading cylinders. The reason for the latter location was the presence prior to flexing of a significant concentration of whitish features which served as stress concentrators and along which the failure ultimately occurred (similar to Figure 4)"*.

More of an aside, the authors may be interested in an observation in the book by D. Masterson published in 2019, "The Story of Offshore Arctic Engineering". It mentions experience in the field of moving a lightly loaded vehicle back and forth on a floating ice road before moving a greater load along it, as a means of improving the load bearing capacity of the ice road. Your work on cyclic flexural loading seems to provide an explanation for this field experience.

We added the following sentence (lines 77-79): "The strengthening of ice is of more than scientific interest, reflected, perhaps in an interesting comment of an arctic engineer who reported that builders of ice roads never trust the ice until it had been "worked in" (Masterson, 2018)".

**Specific Comments**:

Line 21; suggest adding 'and failure' to the end of the sentence.

We added *"and failure"* as suggested.

Line 35; the sentence starting with 'For instance. . .' is not clear, are you saying that the structure is being fatigued, or the ice?

We modified this sentence by adding "or damaged", in the following way: *"For instance, during ice-structure interactions the structure itself, such as a light-house, may be weakened **or damaged** to a degree that depends on the strength of the ice"*. By this sentence (and paragraph) we mean that arctic infrastructure may be susceptible to ice loads and potentially can be damaged. The degree of the damage would depend on the ice properties that are affected by cyclic loading. We did not mean to discuss fatigue of the structure in this sentence.

Line 70; you mention 'recovery', what is being recovered and does that mean increasing or decreasing strength?

We modified the sentence by indicating that strength is being recovered: "*In those experiments, it was discovered that the ice flexural strength increases upon repetitive loading, followed by the recovery of the cyclic-induced increment in strength to the original non-cycled strength upon post-cycling annealing*".

Line 78; Your experiments were performed on and analysed as beams, change 'plates' to beams.

We replaced the word "plate" with "beam" through the text.

Lines 80-90; You mention that ice plates were grown in a circular tank, how deep was the tank? Did you seed the sheet? You mention melt water salinities, were the values given from the tested beams or samples from the whole ice sheet? Density of the ice beams should also be provided, that would help distinguish between brine pockets and air filled voids.

We added that the volume of the tank is 800 L, so it is possible to estimate depth knowing that the diameter is 1 m. We also added the following **(in bold type)** information about the procedure of growing the ice in the tank: *"Briefly, solutions containing 17.5 ± 0.2 ppt and 35 ± 0.2 ppt (parts per thousand, or ‰) of the commercial product "Instant Ocean" salt mixture were prepared and then frozen unidirectionally downward over a period of about 7 days **by using a top-placed cold plate maintained at T = –20±0.1 °C. Before bringing the cold plate into contact with the salt-water solution, the top surface of the solution was seeded with freshwater ice grains of ~ 0.3-1 mm diameter"**. Melt-water salinities mentioned above and also listed in Table 1 are salinities of the ice specimens themselves and not the salinity of the parent ice plate. Densities of the specimens for both low salinity and high salinity ice are also provided in Table 1.*

Line 121; Make clear the orientation of these blocks in the original sheet, presumably the long dimension was in plane of the plate.

We added the following information **(in bold type)** to clarify the orientation of ice blocks: *"Once the ice had been grown, it was cut into blocks of dimensions ~ 10 x 30 x 20 cm³, **where the longest and the shortest dimensions are in the horizontal plane of the original grown ice puck, perpendicular to the direction of growth"**.*

Line 132, Figure 5; This figure indicates that deflection at the centre-point of the beam was measured with respect to the outer pair of loading cylinders of the four-point loading apparatus, this introduces an error. You should be measuring the deflection of the beam with respect to the inner pair of loading cylinders.

We note out that in our test-setup the outer pair cylinders are attached to an immobile upper part of the apparatus (and therefore do not move). The inner-pair of cylinders are connected to the actuator which moves up and down. During cycling, we obtain measurements of the **deflection of the center-point of ice with respect to immobile outer cylinders (LVDT)**. Therefore, we disagree that we introduced an error in our measurements. According to ASTM (Standard Test Methods for Flexural Properties of Materials by Four-Point Bending, D790-17 or D6272-17 for example), "Deflectometer shall be essentially free of inertia at the specified speed of testing. Deflectometer shall be in contact with the specimen at the center of the support span, the gauge being mounted stationary relative to the specimen supports". In our experiments, the deflectometer (LVDT) was indeed attached to the immobile part and was free of inertia.

Line 151; 'softening', what do you mean? Rewrite sentence to be clearer.

We re-wrote this sentence in the following way: *"Measurements of load and of displacement versus time at the beginning and near the end of cycling revealed no evidence of softening (according to Bažant et al. (1984) softening is a decline of uniaxial stress at increasing strain or, in our case, an increase of strain during cycling at constant stress amplitude) during the tests, Figure 6, similar to the case for freshwater ice (Iliescu et al., 2017; Murdza et al., 2020c)."*.

Line 168; were the test beams always in the same orientation as in the original puck? Also for the simple flexural tests, or the final loading to failure after Type 1 or 2 cycling, was the top or bottom surface the one in tension?

Yes, the test beams were prepared and tested always in the same orientation. For example, in line 132 it is stated that thickness dimension of the specimens was parallel to the long axis of the columnar grains.

Specimen thickness is ~1.6 cm, while the thickness of a grown ice puck is ~30 cm; therefore, the thickness of the ice specimens is negligible when compared with the thickness of the parent ice puck. In addition, during specimen preparation (milling) both the top and the bottom surfaces of the ice specimens were prepared in the same manner. Thus, there is no difference whether top or bottom surface of the specimen was the one in tension given the ice properties do not change significantly over the specimen thickness.

Line 186; sp. 'contain'

We corrected this typo.

Line 289; would brine not fill cracks making them difficult, if not impossible, to detect visually; also if I understand the orientation of the thin section, the chance of having a crack in it would be rare.

We agree with this comment and thus added the following sentence to Section 3.6, Acoustic Emissions: *"The reason that microcracks were not observed under the optical microscope may be because they immediately filled up with liquid brine upon formation which results in a loss of contrast".*

As stated in the text, the plane of the thin sections was parallel to the long axis of the columnar grains and parallel to the direction of the greater normal stress (or long axis of the ice beams). With this orientation, we captured the top, the middle and the bottom of the specimen which was cycled; hence, if any crack occurred, it would have initiated most likely either at the top or bottom part of the thin section since those parts correspond to regions with maximum tensile stresses during cycling. Therefore, we think that we chose the correct plane of the thin sections to search for cracks . Indeed, according to the classical fatigue behavior of materials (metals for example), defects (such as brine pockets in our case) would serve as sites for crack initiation

and growth. Therefore, the method that we used would have indicated crack growth/propagation had it occurred.

Line 300; could emissions also originate from grain boundary movements?

Please note that on this point we only summarized the conclusions that other authors presented, not our thoughts here. The previous authors did not discuss whether emission can also originate from grain boundary movements. However, we agree that in general acoustic emission can originate from grain boundary movements (which we also mentioned in our analysis, lines 370-371).

Line 317; 'water hammer' is usually associated with pressure waves in a fluid in a closed system, If you are proposing brine movements in pores, some further explanation of the mechanism is needed.

According to the comments of Reviewer 1 and our further thoughts on this problem, we modified this section and explained the obtained acoustic emission results differently. Please, see below:

There are four possible sources of the noise detected. One is from microcracking. We imagine that microcracks form in regions of mechanical weakness which results in accumulation of damage that we detected via the AE method. Specifically, the brine drainage whitish features discussed above in the test specimens constitute regions of high porosity and thus provide favorable sites for the concentration of such damage. Failure may occur when one of these sites can no longer support the applied stress and a microcrack emerges from the damage zone and propagates. It is possible that newly formed microcracks are stable until a critical length is reached (Cannon et al., 1990; Schulson et al., 1991), at which point the crack growth ensues. The reason that microcracks were not observed under the optical microscope may be because they filled up with liquid brine upon formation which results in a loss of contrast. A second possible explanation for the acoustic emissions is the motion and friction of very fine particles of ice which may have been entrapped inside brine drainage features, as mentioned above. A third possibility is microcracking along grain boundaries due to grain boundary sliding (Elvin and Shyam Sunder, 1996; Goldsby and Kohlstedt, 1997; Mulmule and Dempsey, 1997; Schulson et al., 1997; Weiss and Schulson, 2000). The fourth possible explanation—consistent with the non-history dependence of the hit rate (new Figure 13)-- is a kind of water-hammer effect in which brine entrapped within pockets impacts the wall, first in one direction and then another. None of these possibilities can be evaluated based upon the limits of the present observations. We refrain, therefore, from further speculation on this point.

Line 331; where is the air and brine distributed, separate pockets or both in the same pocket?

We believe that air can be located separately from brine as well as exist as a mixture of brine and air.

Line 347; explain how a brine pocket or channel makes saline ice more susceptible to premature failure. Brine pockets are very rounded, have a much larger radius and lower stress concentration than for a crack.

When we state that saline is more susceptible to premature failure, we compare it with a study on freshwater ice. Indeed, in the study on freshwater ice (Murdza and others, 2020; Iliescu and others, 2017) we did not detect by the unaided eye any defects. Saline ice, however, has a lot of defects when compared with freshwater ice, although these defects, perhaps, are not as "sharp" as cracks, as pointed out by the reviewer. However, although these defects are not as "sharp", significantly more saline specimens failed prematurely during cycling when compared with the study on freshwater ice. This observation is not surprising since it is a well-known fact that porous materials are generally weaker and small cracks grow from the pores, even if pores are spherical (for example, C.G. Sammis and M.F. Ashby (1986) "The Failure of Brittle Porous Solids Under Compressive Stress States").

Line 352; this discussion of brine pockets is very subjective, careful thin sectioning could have provided more definitive information on the grain structure.

We agree that the comparison of interconnected whitish features between ice of lower and higher salinities is subjective. However, we believe that we conducted proper thin section analyses (Figure 1 in the manuscript). Based on the obtained results, there was no significant difference in grain structure in two types of ice (higher and lower salinities). The difference was rather visual which we indicate in Figure 3. This difference was also reflected in the measurements of ice salinity.

Line 376; the tests were done on beams, don't refer to them as plates.

We have made changes through the text.

Line 380; does this conclusion apply to all ice, fresh water and saline?

According to the results we obtained, it seems that the increase in flexural strength upon cycling (beyond a certain number of cycles) scales linearly with the amplitude of the applied outer-fiber stress for both laboratory-grown freshwater ice and laboratory-grown saline ice. A similar effect was also observed in the experiments on natural lake ice (Murdza, Marchenko, Schulson, Renshaw, 2021).

Line 581, Fig. 12; why a negative value of hit rate, start the ordinate at 0.0, also state the units for hit rate on this axis.

We made these corrections.

---

## Author Response (AR2)

**Responses to comments by reviewers of manuscript tc-2020-300 "Behavior of Saline Ice under Cyclic Flexural Loading"**

We thank anonymous referees for additional comments on our work. We have modified our manuscript according to them. Please, see all the responses in red.

**Comments from Referee # 2**

Line 140; it is stated that "loading under constant displacement rate". Was control based on the actuator movement? That would mean that the displacement transducer (LVDT) was being used for measurement, not control. Also I presume reversal of the actuator movement direction was based on a force measurement limit, since report cycling between stress limits. This may have be described in your previous papers, but a sentence or two explaining this here would add to the understanding of the experimental method.

We added the following sentence: "The hydraulic actuator was driven up and down under displacement control with the load limited in both directions".

Line 156; Equation (1) is correct for loading at quarter-points, that is the centre span is half the distance between the outer loading points. While this may be the usual case it is good to be specific.

We added the following (lines 154-155): "the loading span is ½ of the support span". This can also be seen in Figure 5 where we provide all of the dimensions of our 4-point bending apparatus.